# Government Reactions, Citizens’ Responses, and COVID-19 around the World

**DOI:** 10.3390/ijerph19095667

**Published:** 2022-05-06

**Authors:** Jon Reiersen, Manuel Romero-Hernández, Romén Adán-González

**Affiliations:** 1USN School of Business, Department of Business, History and Social Sciences, University of South-Eastern Norway, 3679 Borre, Norway; jon.reiersen@usn.no; 2Department of Applied Economic Analysis, University of Las Palmas de Gran Canaria, 35001 Las Palmas de Gran Canaria, Spain; 3Independent Researcher, 38001 Santa Cruz de Tenerife, Spain; romenadangz@gmail.com

**Keywords:** governance, trust, COVID-19, political culture

## Abstract

We analyze the relationship between different dimensions of the quality of the political system and the outcome of the COVID-19 pandemic. Data are retrieved from open-access databases for 98 countries. We apply a multivariable regression model to identify the relationship between various factors likely to affect the number of COVID-19 deaths, in addition to different dimensions of the quality of the political system. We find that the high quality of the electoral process in a country is associated with more COVID-19 deaths, while good political culture is associated with fewer deaths. As expected, we also find that trust in government and experiences with pandemics in the past is negatively related to COVID-19 deaths. Finally, a high GDP per capita is significantly associated with more COVID-19 deaths. Our findings illustrate that rapid, effective, and comprehensive government measures can protect society from the spread of a virus, but citizen compliance is also essential to policy success.

## 1. Introduction

As of 5 February 2022, more than 5.7 million people have died from COVID-19 worldwide. No country is unaffected by the pandemic, but there are large variations between regions and countries in terms of both the number of the infected and the number of deaths. While Europe has 2187 confirmed COVID-19 deaths per million people, the corresponding figure in Asia is only 279. There are also large differences between countries within different regions of the world. In South America, with an average of 2820 deaths per million people, Peru has 6201 COVID-19 deaths per million compared to 1900 in Uruguay. In Europe, Italy has 2460 COVID-19 deaths per million people compared to 268 in Norway (https://covid19.who.int, accessed on 5 February 2022). What explains the huge differences in the number of COVID-19 deaths across countries? Why have some countries been so much more successful than others in suppressing and controlling the pandemic?

This paper addresses how different dimensions of the quality of the political environment and citizens’ responses to pandemic policies impact COVID-19 outcomes. We know from previous research that the successful management of an acute crisis—such as a pandemic—depends on a complex interplay between many factors [1]. Effective crisis management requires analytical and operational expertise at both local and central levels on the one hand, and capabilities and resources on the other hand. Countries around the world have implemented very different strategies and measures aimed at protecting the society from COVID-19. Variations in the governments’ response do not, however, fully explain differences between countries when it comes to protecting society from the pandemic. Citizen’s compliance is essential to policy success. Human behavior critically affects how a pandemic develops, and well-founded and effective government strategies have little value if the inhabitants still do not believe in them and do not follow them [2,3,4].

The governments’ response to COVID-19 has varied throughout the world, but so has the response of citizens [5,6,7,8,9]. In some countries, people loyally followed the government’s recommendations and measures to control the pandemic, while in other countries, the governments’ policies were met with social unrest and opposition. In still other countries, it was the citizens who asked for effective measures while the governments remained passive or unable to design such policy. The lesson is that a pandemic cannot be fought by the government or civil society alone, and recent experience in different countries shows that the COVID-19 pandemic is not just a health crisis: it has put to the test the cooperation between citizens and governments. The challenging interdependence between effective policies and human behavior that a successful fight against a pandemic requires is the focus of this article.

At the individual level, we know that those who live and work in environments where social distancing is difficult to maintain have a higher risk of being infected by COVID-19. Age is also strongly associated with the likelihood of becoming seriously ill because of COVID-19. Those with low incomes are also more at risk, probably because those with low incomes have living conditions and jobs where they live and work close to others. Low income also limits the opportunities to take precautions that can protect against infection, such as working from home. Likewise, people with diabetes and obesity have a higher COVID-19 mortality rate [10].

To understand the differences between countries in terms of the number of COVID-19 deaths, it is necessary to look beyond differences at the individual level. Countries have responded to COVID-19 with various policy measures, but their design, rigorousness, and reach varied significantly. As [11,12,13] show, the timing and reach of the interventions can explain a lot of the differences between countries in fighting the pandemic. Ref. [12] compare five East Asian countries—China, South Korea, Taiwan, Hong Kong, and Japan—with six Western countries—France, Germany, Italy, Spain, the United Kingdom, and the United States—showing that the six Western countries have experienced 41 times more COVID-19 deaths per million people than the five East Asian countries. The five East Asian countries have an unweighted average of 5552 COVID-19 deaths per million population, compared to 228,681 in the six western countries. This is a substantial difference. Ref. [12] conclude that the difference between the two regions can largely be explained by an earlier and stronger government response. An important part of the East Asian strategy was a rapid government response to local virus outbreaks and a strong priority of bringing infection rates down to zero. This was combined with more stringent mobility control as well as more comprehensive testing, tracing, and isolation policies already in the very early stages of the pandemic. In contrast, Western countries (France, Germany, Italy, Spain, the United Kingdom, and the United States) preferred to keep their society open, even when they saw that infection rates were rising rapidly. Hesitant government responses in these countries in the early stages of the pandemic caused many people to become infected.

Developing effective government strategies that can protect society from the spread of a virus is crucial, but as noted, how the population reacts and responds to these strategies is just as important. A pandemic can be viewed as a classic collective action problem where there is a conflict between what serves the individual and what serves the common good [14]. If people generally trust the government and the advice it gives, if this advice communicated in a consistent manner, and if people tend to follow it (take precautions, adopt good hygienic routines, keep physical distance and so on), a pandemic can be brought under control—which is in everyone’s best interest. However, if people generally believe that everyone is doing what it takes to bring the pandemic under control, it is tempting to free ride on the joint efforts of others. Free-riding saves the individual from the cost of taking precautions at the same time as the pandemic is brought under control as a result of the efforts of others—but, of course, if everyone freerides, the virus spreads and the society faces a pandemic. Solving collective action problems require cooperation and a key source of cooperative behavior is what is often labeled social capital. A high level of social capital in society, defined as shared social norms, trust, and effective social sanctioning systems that punish free-riding and other forms of anti-social behavior, is generally assumed to contribute to more socially responsible behavior [15,16,17,18]. It is clear that a social norm such as “You should not take advantage of your fellow citizens by shirking on the necessary actions needed to secure the common good” is a valuable asset for a society in the face of a pandemic.

Up to now, authoritarian countries seem to have managed to control the COVID-19 pandemic better than democratic countries. As [19] note: “(…) democratic countries’ COVID-19 death rate is on average larger than that of non-democratic countries by approximately 42 per 100,000 inhabitants. That is, the fatality rate in a democracy is on average 3.7 times larger than in an autocracy”. Such numbers have led many to question whether more authoritarian countries have an edge over democratic countries in pandemic response. Autocratic governments may reduce the problem of socially irresponsible behavior by using force, and they may also be quicker in mobilizing the necessary resources without considering democratic processes. Recent research has shown that political factors indeed have influenced COVID-19 responses. While some studies find that countries with a higher democracy level were relatively slower to implement COVID-19 measures, and have suffered from higher COVID-19 infection and death rates, other studies report more mixed findings [19,20,21,22,23,24].

We also use “the level of democracy” as an independent variable in our study, but we do it by utilizing a dataset that has been little used in the COVID-19 health outcome research. Our measure of democracy is taken from the Economist Intelligence Unit’s Democracy Index 2020 (https://www.eiu.com/n/campaigns/democracy-index-2020/, accessed on 1 December 2021) making it possible to disentangle two distinct dimensions of democracy. The first dimension, electoral process, and pluralism captures the more standard and minimalistic measure of the level of democracy in a country: to what extent are elections free and fair, can citizens cast their vote free of threats to their security, are citizens free to form political parties, and so on (EIU, 2020). The second dimension of democracy, democratic political culture, captures the deeper structures of democracy: how strong is the citizens’ support for democracy as a form of government? Is there a sufficient degree of societal consensus to reinforce a stable democracy—or is the society characterized by conflict and polarization, and a desire for a strong leader who bypasses parliament and elections to get things done? How prepared are politicians, experts, and citizens to stand together to solve the collective action problem and the social crisis that a pandemic represents?

The aim of this paper is to address how different dimensions of the quality of the political environment impact COVID-19 death rates across countries. We have collected data of per capita death rates for 98 countries around the world (see Table A1 in the Appendix A), in addition to different variables that capture various dimensions of the social and political context in each country. These variables are discussed in more detail in the next section, together with a presentation of the statistical analysis used.

## 2. Materials and Methods

Table 1 summarizes the descriptive statistics of the variables included in our analysis. Our dependent variable is the confirmed cumulative COVID-19 deaths per million people measured by the first week of November 2021 and retrieved from Our World in Data, Oxford University (https://ourworldindata.org/, accessed on 1 December 2022).

Political system quality has been measured by taking the Electoral Process and Pluralism Index (EP) and the Political Culture Index (PC) from The Economist Intelligence Unit of Democracy Index report [25], the Electoral Process and Pluralism Index (EP), and the Political Culture Index (PC). These two indexes are based on the sum of several indicator scores within each category, converted to a 0 to 10 scale.

As already noted, the EP index captures the more standard and minimalistic measure of the level of democracy: to what extent are elections free and fair? Are citizens free to form civic political organizations independent of the government? Do opposition parties have a realistic chance of forming the government [25]? The PC index captures deeper dimensions of democracy. This index is based on answers to eight questions designed to measure how the respondents view democracy as a political system and their belief in its legitimacy.

Previous research has shown that trust is a critical factor in addressing a crisis like COVID-19. Individuals who trust the government are more willing to comply with the government’s infection control measures, and a country with more trusting individuals is probably better prepared to fight a pandemic, all else equal [4,5,6,26,27].

We have included “Trust in Government” (TG) as an explanatory variable in our study and collected the data from the indicator included in the Wellcome Global Monitor 2018 [28]. We collected the information from the question posed in the yearly global survey: “How much do you trust the national government in this country?” We took the share of respondents who answered with the alternatives “A lot” or “Some”. We have also included two control variables, gross domestic product per capita (GDP), measured in constant 2019 international dollars, and previous experience with pandemics. This last variable includes the list of countries that had previously experienced with a corona pandemic [13].

Table 1 summarizes the descriptive statistics of the variables included in our study, while Figure 1, Figure 2, Figure 3 and Figure 4 illustrate the bivariate relationship between the different variables and COVID-19 deaths (see Figure A1 in Appendix A for the univariate distributions plot).

## 3. Empirical Results

We estimate a multivariable regression model to investigate the relationship between the COVID-19 death level and the explanatory variables. The regression equation for per capita COVID-19 deaths is as follows:$$D_{i}\left(7\;\mathrm{November}\;2021\right)=\propto +\beta \left(EP_{i}\right)+\gamma \left(PC_{i}\right)+\delta \left(TG_{i}\right)+\varepsilon \left(\log(GDP_{i})\right)+\theta \left(EXP_{i}\right)$$
where *i* indexes a country, *D* denotes COVID-19 deaths per million people, *EP* stands for electoral process and pluralism, *PC* stands for political culture, *TG* is trust in government, *GDP* is gross domestic product per capita, and *EXP* denotes previous experience with pandemics.

The empirical model was estimated using OLS. The estimated parameters of our regression model are in Table 2. Most of the parameters are statistically significant at the 0.05 level. The R^2^ value for the estimated model is 0.51 and indicates that the data fits the model well.

The focus of this paper is to identify how both social and political behavior affects combating the COVID-19 pandemic. In Table 2, the first parameter shows how social behavior has had a determinant influence in the expansion of the virus. The negative sign of the trust in government estimator shows that societies with a higher level of trust in the government and their policies have been able to contain more the number of deaths, in line with other results obtained by [29]. This means that when the trust in the government increases by one unit, the average estimated covid deaths per million is reduced by 1.51.

The negative sign for the political culture estimator reveals how more advanced societies with higher political capital (see [30]) and governments with more knowledge and experience will contain the virus better. Political culture refers to how society develops and how citizens can be identified with their political system.

In a pandemic situation, behaving under the conception of community, it is determinant to keep physical distance and follow the recommendations in order of the common interest. Societies with lower political culture have more of a tendency for individualism, and free riders appear, making easier the expansion of the virus. Regarding electoral process and pluralism, the positive and significant relationship between this variable and our dependent one means that societies with a higher level of representativeness and chambers closer to the map of citizens’ preferences will have more difficulties containing the virus. Ref. [31] have similar results. Ref. [21] Cepaluni (2020) found that a higher level of democracy raises deaths. Both Annaka et al. (2021) [20] and Cassan et al. (2021) [19] did not see their political influence on fatalities. Ref. [32] had found a positive correlation between democracy and health, but not under pandemic conditions.

These more pluralist and more experienced chambers would struggle more to adopt stricter quarantines and, therefore, limit fewer people’s movements. In a pandemic, it is expected that it would be easier to implement public policies to limit citizens’ liberties in an autocratic regime. This makes it easier to contain the virus, for example, forcing people to be confined even if they are not infected. These low-cost policies will be less costly for a government as soon as the electoral process is less transparent and representative.

The positive sign of the logarithm of the GDP-per capita estimator (see also for similar results [21] shows that more prosperous societies, with a higher level of economic activity, are the perfect camp for a virus. More economic activity means more interactions between citizens, more international flights, denser, and bigger cities. Even with better health resources, the limited knowledge about the disease makes these societies more vulnerable. The population in more prosperous countries also generally has a longer life expectancy, which for the covid disease it is not a positive vector as the treatments are still not fully developed enough to avoid deaths when older people get infected.

We hypothesized that countries with previous experience in managing similar pandemics would have a lower number of deaths. Although the coefficient is not significant at 95%, the sign is as expected and does acquire significance at 90%. Given that we are working with a small sample (only five countries are classified as countries with prior epidemic experience), it is plausible to expect that with a larger sample the level of significance would have been stronger. Hence, our results indicate that countries with prior experience in combating pandemics have also been more successful in combating the COVID-19 pandemic.

The direction of the effect is essential, (the sign of the coefficient), and which factor is the most fundamental. For this purpose, we again implemented the regression with the standardized variables, which means that all variables are now in the same unit (standard deviations). The results are in Table 3. Although all variables have a significant relationship with the dependent variable, not all have the same impact. The political culture and the (log) GDP are those with the more potent effect, followed by the electoral process and pluralism variable. All the factors can hardly be modified in the short term when facing a pandemic. Nevertheless, except in the case of experience, the effect size of the variables is quite similar, which shows how complex it is to design and deploy the best political action.

## 4. Discussion

We have found evidence that the number of deaths caused by the COVID-19 pandemic can be explained by a group of factors. First, we found that countries with previous pandemic experience were able to better manage the expansion of the virus and the sickness resulting in a smaller number of deaths. Using the consideration of the virus as a public good influenced how the citizens behaved with other members of society following the recommendations of the authorities, maintaining social distance, and avoiding contact which was determinant first in the spreading of the virus and consequently in the number of deaths. The model shows that when citizens place greater trust in their government the expansion of the virus is controlled, and the number of deaths falls.

Governments or citizens alone cannot individually manage the deaths and the spread of disease generated by a virus such as COVID-19, they must act in unison. Governments must provide recommendations, tracking, healthcare services, and vaccination, while citizens must behave according to the general interest instead of thinking of themselves, collaborating, and following all recommended measures of prevention.

A higher political culture also means that the government accumulated an increased knowledge of good public policies. This means that in the past this government was able to provide better public health care services in both primary and clinical areas. This knowledge and efficiency would also be converted to improved vaccination measures. All of these factors allow governments to better manage COVID-19 sickness and the tracking and avoidance of deaths.

Management of a pandemic is linked to political capital. Prior to the outbreak of COVID-19, successful management was highly related to good public policies adopted in the past linked to a large investment in both primary and clinical healthcare resources. The first resource is evident in the tracking of the infection and the second when treating the disease. Other types of governments that did not invest in healthcare in the past have been forced to put the population in lockdowns, which is an inexpensive way of avoiding the effects of the pandemic in the short run, but a more costly solution in the mid-term and long-term in terms of economic activity and inequality. This approach of avoiding the disease and deaths cannot be sustained for a long time because both people and the economy cannot be kept in a permanent lockdown. This is especially true, in the midst of the sixth wave of the pandemic at the moment that this paper is being written (January 2022).

The Spanish government has been reducing the healthcare system budget for almost a decade while lengthening public service healthcare waiting lists and transferring many treatments and medical interventions to the private sector [33]. The government response in March 2020 was one of the hardest in Europe: during the first two months of the pandemic, people were in complete lockdown and could only leave their homes for a period of one hour and limited their travel to 5 km. Police and military were on alert to maintain order in the streets and enforce home lockdown, even prohibiting walks in the mountains or any outdoor area. Hospitals were completely saturated, and many measures of the health authorities were erratic. At that time, Spain had the highest number of deaths per COVID-19 in the world. The Spanish government also declared a state of emergency and limited the functions of the parliament (https://www.lamoncloa.gob.es, accessed on 21 April 2022). Spain has a short democratic history with a limited electoral system and chamber/congress traditionally dominated by two political parties. Later in 2021, the highest court in Spain declared these adopted measures to be unconstitutional because they limited civil liberties.

Societies that are considered to possess a high quality of democracy with better and more representative electoral systems are closer to the preferences of citizens. Governments that have adopted less drastic measures to restrict civil liberties allowing greater freedom of movement for their citizens have led to a higher level of contact and consequently more infections and later deaths. A similar effect occurs with a pluralistic chamber. Then governments have had more difficulties to approve faster measures and more unilateral with regards to maintaining civil liberties.

Plurality in parliament causes delays and hampers governments from adopting drastic measures. Citizens have greater freedom of movement and therefore, infections and deaths rise, as our model has shown. Every state within Spain is responsible for managing the pandemic within its territory in the sixth wave. The Catalonian government is a state with a low plurality and an electoral system with some deficiencies which has a poor provision of public services, and the first one which approved the lockdown the first after middle night in the country during the sixth wave.

Sound policies adopted in the past have also led to a high level of trust of citizens in their governments, which is necessary for them to follow their recommendations and be successful in the management of the pandemic [29]. High political culture is also related to greater integration of citizens in the society, leading to appropriate behavior which is necessary in a pandemic and can be considered a public good.

How governments manage the pandemic will affect the behavior of citizens. Established good policies in the present will also result in maintaining the citizens’ trust and increase their involvement in the control of the virus, following recommendations and looking out for the general interest. In countries with a higher political culture, citizens are more active and integrated into society. Civil activism led people to be closer to the feeling of community which in a pandemic is determinant. The virus can be considered a public good where the existence of free riders looking for their individual interest can destroy the outcomes achieved by the rest of the community.

The COVID-19 pandemic has not only created a healthcare crisis but an economic one as well, shedding light on what was wrong in both social and political systems. In many countries, the pandemic has resulted in a reduction of civil liberties and therefore a deterioration of democracy [25]. It has also shown the weakness of governments that were hidden by strong economies and populist political messages. In this type of countries citizens have not only suffered the impact of the virus in deaths but also in the loss of civil rights. Starting with lockdown, now governments are pushing to make vaccinations obligatory for citizens while continuing to lose rights and liberties, even in countries which are considered established democracies. The pandemic has also shown that the more successful societies are those with a strong citizen integration and therefore with higher social capital. Both the welfare and the goals of a society are always the result of the participation of the public sector but also involve the private sector and the civil society.

Finally, the model shows that more developed economies present a higher level of deaths from the COVID-19 sickness. More economic activity means more interactions between the population, denser cities, more international flights, and more difficulties to stop interactions. All of this occurs, even if richer countries will always have better healthcare systems, and because COVID-19 treatment still has not been developed, which has resulted in more deaths.

## 5. Conclusions

We have analyzed the relationship between different dimensions of the quality of the political system and the outcome of the COVID-19 pandemic with data for 98 countries. Using a multivariable regression model, we find that the high quality of the electoral process in a country is associated with more COVID-19 deaths while good political culture is associated with fewer deaths. As expected, we also find that trust in government and experiences with pandemics in the past is negatively related to COVID-19 deaths. Finally, a high GDP per capita is significantly associated with more COVID-19 deaths.

## Figures and Tables

**Figure 1 ijerph-19-05667-f001:**
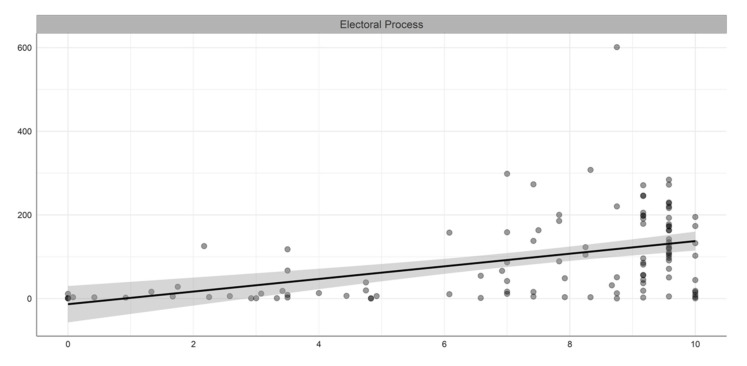
Bivariate relation between electoral process and COVID-19 deaths per million people.

**Figure 2 ijerph-19-05667-f002:**
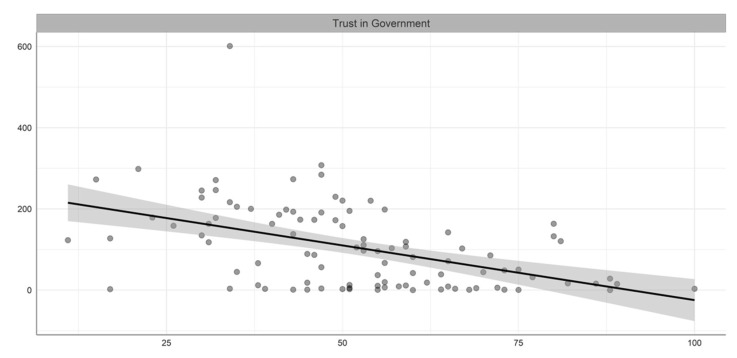
Bivariate relation between trust in Government and COVID-19 deaths per million people.

**Figure 3 ijerph-19-05667-f003:**
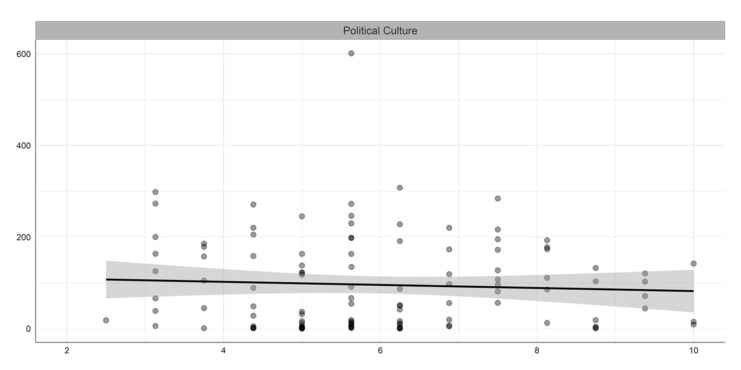
Bivariate relation between political culture and COVID-19 deaths per million people.

**Figure 4 ijerph-19-05667-f004:**
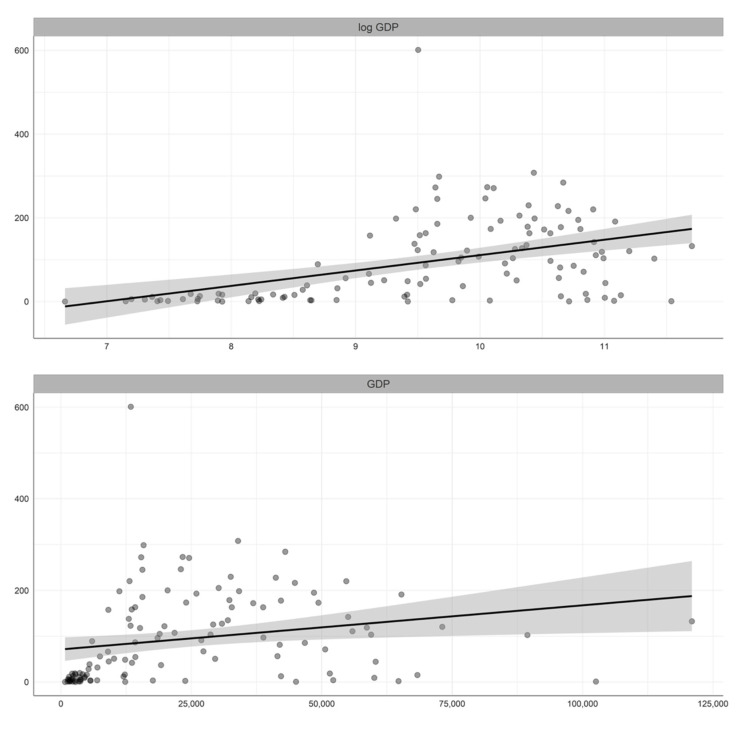
Bivariate relation between GDP and COVID-19 deaths per million people.

**Table 1 ijerph-19-05667-t001:** Summary statistics for the sample variables.

Variable	Std. Dev.	Min	Pctl. 25	Pctl. 75	Max	Source
Trust in Government	17.22	11	40.75	64.25	89	Wellcome Global Monitor
Electoral Process	2.27	0.42	7	9.58	10	The Economist Intelligence Unit
Political Culture	1.84	3.13	5	7.5	10	The Economist Intelligence Unit
GDP per capita	23.90	1.28	11.8	42.39	12.09	World Bank
log. GDP per capita	1.11	7.15	9.38	10.66	11.70	

**Table 2 ijerph-19-05667-t002:** Estimated Model.

Predictors	CD Estimates
Intercept	−78.293
	(72.186)
Trust in Government	−1.589 *
	(0.494)
Electoral Process and Pluralism	12.388 *
	(3.940)
Political Culture	−21.865 *
	(5.441)
Gross domestic product (log GDP)	32.122 *
	(8.901)
Experience	−61.111 ^+^
	(35.167)
Observations	98
R^2^	0.511
Adjusted R^2^	0.485
Residual Std. Error	73.546 (df = 92)
F Statistic	19.251 (df = 5; 92)

Note: ^+^ *p* < 0.1; * *p* < 0.01.

**Table 3 ijerph-19-05667-t003:** Standardized coefficients.

Predictors	Estimates	Standardized
Trust in Government	−1.59	−0.27
Electoral Process and Pluralism	12.38	0.34
Political Culture	−21.87	−0.39
Gross domestic product (log *GDP*)	32.12	0.38
Experience	−61.11	−0.13

## Data Availability

Data available on request due to restrictions.

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
