# Peer review of "Government Reactions, Citizens’ Responses, and COVID-19 around the World"

_ijerph, 2022, doi:10.3390/ijerph19095667_

Round 1
Reviewer 1 Report
23-30;Please add citations.
269; “sixth wave of the pandemic at the moment that this paper is being written”; Please specify the period.
272-283; Add citations as they contain objective facts.
317-349; The scientific conclusion is inaccurate. The original text should be included in the discussion. Instead, replace it with the following sentences and conclude.
“We analyse the relationship between different dimensions of the quality of the political 9 system and the outcome of the COVID-19 pandemic. Data are retrieved from open access databases 10 for 98 countries. We apply a multivariable regression model to identify the relationship between 11 various factors likely to affect the number of COVID-19 deaths, in addition to different dimensions 12 of the quality of the political system. We find that high quality of the electoral process in a country 13 is associated with more COVID-19 deaths, while good political culture is associated with fewer 14 deaths. As expected, we also find that trust in government and experiences with pandemics in the 15 past is negatively related to COVID-19 deaths. Finally, a high GDP per capita is significantly associated with more COVID-19 deaths.”
Author Response
Thank you very much for your comment to the article we honestly think the paper has improved with them. We have tried to adopt all of them.
23-30;Please add citations. Done
269; “sixth wave of the pandemic at the moment that this paper is being written”; Please specify the period. Line 287. Added January 2022.
272-283; Add citations as they contain objective facts.
Line 301. Added official reference of the Spanish government https://www.lamoncloa.gob.es
Line 292. Added reference Bandres and Gonzalez (2015)
317-349; The scientific conclusion is inaccurate. The original text should be included in the discussion. Instead, replace it with the following sentences and conclude.
“We analyse the relationship between different dimensions of the quality of the political 9 system and the outcome of the COVID-19 pandemic. Data are retrieved from open access databases 10 for 98 countries. We apply a multivariable regression model to identify the relationship between 11 various factors likely to affect the number of COVID-19 deaths, in addition to different dimensions 12 of the quality of the political system. We find that high quality of the electoral process in a country 13 is associated with more COVID-19 deaths, while good political culture is associated with fewer 14 deaths. As expected, we also find that trust in government and experiences with pandemics in the 15 past is negatively related to COVID-19 deaths. Finally, a high GDP per capita is significantly associated with more COVID-19 deaths.”
Following this suggestion we have integrated the original text of conclusion in the section discussion (Lines 253-347) and add to conclusion the text suggested by the referee. Line 350-359
Reviewer 2 Report
An interesting paper, but one I feel needs more work. The issue I have with the paper and the analysis within it is presentation of the data: we are told that 98 countries are in the dataset, and then we are told about the general qualities of said data without much information about how fine-grained the analysis is. We are given tables aggregating that data, but not given any tabulated information about the 98 countries and how they score on the various metrics the paper presents as important for understanding various responses to the pandemic. At a minimum we need a table in the article which gives us a rundown of the sampled countries, rather than just the aggregated data. We cannot infer much from the data aggregates.
Let me give an example of the kind of worry I have here: in aggregate the data looks bad for the West and good for Asian countries, and COVID responses look better (if we care about population health outcomes) for authoritarian regimes. Yet two particular examples stand out against this aggregated data: Australia and New Zealand. Both are well-performing democracies with good GDPs, etc., and had equivalent health outcomes to some of the Asian countries.
Now, Australia and New Zealand might be outlier polities but, at the same time, they are also known to have had better-than-average responses to COVID outcomes than other equivalent Western nations, and it bears asking the question "Why?" and can this data explain it? If so, it should be possible to use them as an example. If not, then the fact they aren't mentioned does raise a question as to whether this data actually gives us the right framework.
Some minor comments:
The paper claims:
"If everyone follows the governments' advice, takes precautions, adopt good hygienic routines, keep physical distance and so on (e.i. cooperate), a pandemic can be brought under control, which is in everyone’s best interest."
Surely this depends on the advice and the consistency of the advice: masking advice, for example, varied in Western countries and over time, making it hard for some citizens to understand the right thing to do. As such, good communication is key here, rather than just following advice.
Author Response
Thank you very much for your comment on the article we honestly think the paper has improved with them. We have tried to adopt all of them.
An interesting paper, but one I feel needs more work. The issue I have with the paper and the analysis within it is presentation of the data: we are told that 98 countries are in the dataset, and then we are told about the general qualities of said data without much information about how fine-grained the analysis is. We are given tables aggregating that data, but not given any tabulated information about the 98 countries and how they score on the various metrics the paper presents as important for understanding various responses to the pandemic. At a minimum we need a table in the article which gives us a rundown of the sampled countries, rather than just the aggregated data. We cannot infer much from the data aggregates.
Answer:
Line 141 we have added a reference to Annex to table A1 with the list of countries included in the sample
Figure A1 we have added an univariate distributions plot for the variables included in the model
Let me give an example of the kind of worry I have here: in aggregate the data looks bad for the West and good for Asian countries, and COVID responses look better (if we care about population health outcomes) for authoritarian regimes. Yet two particular examples stand out against this aggregated data: Australia and New Zealand. Both are well-performing democracies with good GDPs, etc., and had equivalent health outcomes to some of the Asian countries.
Now, Australia and New Zealand might be outlier polities but, at the same time, they are also known to have had better-than-average responses to COVID outcomes than other equivalent Western nations, and it bears asking the question "Why?" and can this data explain it? If so, it should be possible to use them as an example. If not, then the fact they aren't mentioned does raise a question as to whether this data gives us the right framework.
Answer:
The model explains the case of Australia and New Zealand since they are included in the dataset. In addition to being well performing democracies they are also included in the group of countries with previous experience with pandemics (see Table A1 in the Annex). This have given them an advantage compared to others western countries with equivalent characteristics.
Some minor comments:
The paper claims:
"If everyone follows the governments' advice, takes precautions, adopt good hygienic routines, keep physical distance and so on (e.i. cooperate), a pandemic can be brought under control, which is in everyone’s best interest."
Surely this depends on the advice and the consistency of the advice: masking advice, for example, varied in Western countries and over time, making it hard for some citizens to understand the right thing to do. As such, good communication is key here, rather than just following advice.
Answer: We have adjusted the claim referred to above.
Reviewer 3 Report
This paper discussed the impact of government and citizens on the control of COVID-19. The question is interesting yet difficult to answer. The authors try to solve this problem using six variables from the perspective of the economy and politics. The model's result seems fittable, and the discussion is describable. However, I think the results and conclusions may not be with enough evidence
1, From the beginning to the end, I do not see any detailed data even in the supporting files. I question the results can be concluded without this data. And it impacts the results and conclusions without specific countries or regions.
2, Is the data used in this study reliable? Every journal or organization can put out an index or figure, but it definitely is subjective and without proper testimony.
3, The authors conclude partial reason of the high death rate of richness centuries to the high activities and more elder people. It's just a distortion of causality
In addition, there some issues should be explained.
1. There are some spelling and abbreviation mistakes.
2. How does the “previous experience with pandemics” measure in this article?
3. Please give the list of 98 countries used in this study.
4. Why just use the death data in the first week of 2021.11
5. How does the EP and PC indicators compute?
Author Response
Thank you very much for your comment on the article we honestly think the paper has improved with them. We have tried to adopt all of them.
This paper discussed the impact of government and citizens on the control of COVID-19. The question is interesting yet difficult to answer. The authors try to solve this problem using six variables from the perspective of the economy and politics. The model's result seems fittable, and the discussion is describable. However, I think the results and conclusions may not be with enough evidence
1, From the beginning to the end, I do not see any detailed data even in the supporting files. I question the results can be concluded without this data. And it impacts the results and conclusions without specific countries or regions.
Line 140. We have added a reference to table A1 in the Annex with the list of countries included in the sample
Line 178. We have added Figure A1 in the Annex with a univariate distributions plot for the variables included in the model
2, Is the data used in this study reliable? Every journal or organization can put out an index or figure, but it definitely is subjective and without proper testimony.
Data were obtained from different sources. Line 178, Table 1 summarizes the different reliable sources for the data.
Table 1. Summary statistics for the sample variables.
|
Variable |
Std. Dev. |
Min |
Pctl. 25 |
Pctl. 75 |
Max |
Source |
|
Trust in Government |
17.22 |
11 |
40.75 |
64.25 |
89 |
Wellcome Global Monitor |
|
Electoral Process |
2.27 |
0.42 |
7 |
9.58 |
10 |
The Economist Intelligence Unit |
|
Political Culture |
1.84 |
3.13 |
5 |
7.5 |
10 |
The Economist Intelligence Unit |
|
GDP per capita |
23.90 |
1.28 |
11.8 |
42.39 |
12.09 |
World Bank |
|
log. GDP per capita |
1.11 |
7.15 |
9.38 |
10.66 |
11.70 |
|
More fully references are given in the article.
Line 147. Covid data deaths were obtained from Our World in Data Oxford University (https://ourworldindata.org/).
Line 150. Political system. The Pluralism Index´ (EP) and the Political Culture Index´ (PC) is taken from The Economist Intelligence Unit of Democracy Index report (EUI, 2020) (https://www.economist.com/)
Line 166 We have included "Trust in Government" (TG) as an explanatory variable in our model. The variable is taken from the Wellcome Global Monitor 2018 (https://wellcome.org/reports/wellcome-global-monitor/2018)
3, The authors conclude partial reason of the high death rate of richness centuries to the high activities and more elder people. It's just a distortion of causality
- In addition, there some issues should be explained.
4.1There are some spelling and abbreviation mistakes.
We have carefully gone through the document again to correct spelling and abbreviation mistakes.
4.2. How does the “previous experience with pandemics” measure in this article?
Line 173. Previous pandemic experience is a dummy variable in the model which refers to countries with past experience in managing major pandemics. We have added the list of these countries in the Annex (Table A1).
4.3.Please give the list of 98 countries used in this study.
We have added the list of countries used in the Annex (Table A1)
4.4.Why just use the death data in the first week of 2021.11
We have used commulative number of Covid-19 deaths up to the first week November 2021. This is the date when we collected the data and started our analyses. The pandemic is still ongoing, but we had to choose a time of reference.
4.5. How does the EP and PC indicators compute?
The Electoral Process and Pluralism Index´ (EP) and the Political Culture Index´ (PC) is taken from the dataset “Democracy Index 2020” constructed by The Economist Intelligence Unit (see The Economist Intelligence Unit of Democracy Index report, 2020 and https://www.economist.com). Both indexes are based on the sum of several indicator scores within each category, converted to a 0 to 10 scale. We have discussed this both in the Introduction and the Material and Methods section. Further information can be found at https://www.eiu.com/n/campaigns/democracy-index-2020/
Reviewer 4 Report
The article is well written, in a concise and clear narrative, throughout the methodology, results and discussion sections.
There are a few details in need of correction:
- The question of the “level of democratic” is addressed in the introduction and methodology. It should only be addressed in the methodology.
- How did are collect the information of the previous experience of a country in a pandemic? It is important to clarify this aspect in the methodology.
Author Response
Thank you very much for your comments on the article we honestly think the paper has improved with them. We have tried to adopt all of them.
The article is well written, in a concise and clear narrative, throughout the methodology, results, and discussion sections.
There are a few details in need of correction:
1.The question of the “level of democratic” is addressed in the introduction and methodology. It should only be addressed in the methodology.
Done
2.How did are collect the information of the previous experience of a country in a pandemic? It is important to clarify this aspect in the methodology.
Line 178. Previous pandemic experience refers to countries which have experiences in managing previous pandemics. These countries are now listed in the Annex (Table A1).
Round 2
Reviewer 2 Report
This is a short review, but in order to make this review I not only had to re-read the paper but also read another paper to check that that authors were, in fact, citing it correctly.
The paper in question is Jeffrey D. Sach's "Comparing COVID-19 Control in the Asia-Pacific and North Atlantic Regions" which the authors claims allows them to claim that Australia and New Zealan did well in the current pandemic due to past pandemic experience. As a citizen of one of these countries I happen to know this claim is false, but I needed to check to see if the problem was with the authors' interpretation of the Sach's article, or whether Sach himself was wrong.
Unfortunately, the problem seems to be a little of both. Let me explain:
Sach's discussion of epidemic preparedness occurs on page 32, where he writes:
"Lack of epidemic preparedness. The Asia-Pacific advantage also seems to reflect epidemic preparedness, probably reflecting the region’s recent intensive battles with SARS, H1N1, MERS, and other zoonotic diseases. These experiences gave rise to effective national emergency response systems and to strategies for regional cooperation, including the WHO Western Pacific Regional Office Asian-Pacific Strategy for Emerging Diseases (APSED), now in its third iteration. The North Atlantic countries were far less experienced with epidemic threats."
We might take it that he means the Asia-Pacific in a general sense (i.e including Australia and New Zealand), in which case he would be wrong (SARS, H1N1, MERS were issues in Asia but not Pacific nations like Australia and New Zealand). However, earlier, when talking about physical distancing, face-mask wearing, and closure of events he states:
"A recurring ideological claim in North Atlantic public debate has been the assertion that NPIs such as physical distancing, face-mask wearing, and closure of events constitute a denial of freedom, thereby limiting the public’s compliance. Such ideological claims seem to be far less frequent in the Asia-Pacific countries, even including Australia and New Zealand, which share the British liberal traditions."
That is, Australia and New Zealand are added in here as exceptions.
The problem here is that Sach sometimes means "Asia-Pacific" as inclusive (adding in Western nations in the region like like Australia and New Zealand), and sometimes he does not. Unfortunately for the authors of the paper under revew, Sach's does not mean that Australia and New Zealand have prior epidemic experience (at least, not since the Spanish Flu epidemic in the early 20th Century). Indeed, had the authors thought to check this, rather than rely on Sach, they would have easily seen that Australia and New Zealand have no prior epidemic experience, and so my original concern is simply not addressed. Thus, let me reiterate it:
""Let me give an example of the kind of worry I have here: in aggregate the data looks bad for the West and good for Asian countries, and COVID responses look better (if we care about population health outcomes) for authoritarian regimes. Yet two particular examples stand out against this aggregated data: Australia and New Zealand. Both are well-performing democracies with good GDPs, etc., and had equivalent health outcomes to some of the Asian countries.
"Now, Australia and New Zealand might be outlier polities but, at the same time, they are also known to have had better-than-average responses to COVID outcomes than other equivalent Western nations, and it bears asking the question "Why?" and can this data explain it? If so, it should be possible to use them as an example. If not, then the fact they aren't mentioned does raise a question as to whether this data actually gives us the right framework."
The authors think this concern can be addressed by reference to Sach. However, it cannot. The concern still stands.
It is also important to note that whilst the Sach's paper was formally published in 2021, the data it refers to reflects the state up to June 2020. Sach does claim the data is consistent throughout the rest of 2020, but given the fast pace and hanging situation, I would expect more recent data to be cited here.
As such, my worry about the paper: the argument needs more work.
Author Response
We really appreciate the detailed comments and the correction about considering Australia and New Zealand as countries with prior epidemic experience. We also appreciate the referee's clarifications of the confusion that may result from Sachs' (2021) discussion.
We have reestimated our empirical model, where we have taken out Australia and New Zealand from the sample of countries with past pandemic experience. Fortunately, the results from this reestimation are in line with our first model (where Australia and New Zealand are included in the sample of countries with past pandemic experience). The variable “Experience” keeps the expected negative sign, and is significant at 90%. Our previous conclusion thus seems to hold: Countries with previous experience in combating pandemics have also been more successful in combating the COVID-19 pandemic.
We have modified lines 246 to 251 to adapt the new sample and results. We have modified Table A1 Countries in the Sample to include Australia and New Zealand as No Past experience countries
Reviewer 3 Report
The authors have made improvements and addressed most of my concerns. However, there are some flaws in this study.
Author Response
We have made some changes in the paper related to Australia and New Zealand as countries with prior epidemic experience following the suggestion of a revieweer.
We have reestimated our empirical model, where we have taken out Australia and New Zealand from the sample of countries with past pandemic experience. Fortunately, the results from this reestimation are in line with our first model (where Australia and New Zealand are included in the sample of countries with past pandemic experience). The variable “Experience” keeps the expected negative sign, and is significant at 90%. Our previous conclusion thus seems to hold: Countries with previous experience in combating pandemics have also been more successful in combating the COVID-19 pandemic.
We have modified lines 246 to 251 to adapt the new sample and results. We have modified Table A1 Countries in the Sample to include Australia and New Zealand as No Past experience countries